# Character Recognition in Endangered Archives: Shui Manuscripts Dataset, Detection and Application Realization

Minli Tang [1,2,3] , Shaomin Xie [1,3], Mu He [1,3] and Xiangrong Liu [1,3,*]

1    Department of Computer Science and Technology, School of Informatics Xiamen University, Xiamen University, Xiamen 361005, China; tangml@stu.xmu.edu.cn (M.T.); xsmin@stu.xmu.edu.cn (S.X.); hemuoj@163.com (M.H.)
2    School of Big Data Engineering, Kaili University, Kaili 556011, China
3    Key Laboratory of Digital Protection and Intelligent Processing of Intangible Cultural Heritage of Fujian and Taiwan, Ministry of Culture and Tourism, Xiamen University, Xiamen 361005, China
*    Correspondence: xrliu@xmu.edu.cn; Tel.: +86-13616048924

**Abstract:** Shui manuscripts provide a historical testimony of the national identity and spirit of the Shui people. In response to the lack of a high-quality Shui manuscripts dataset, we collected Shui manuscript images in the Shui area and used various methods to enhance them. Through our efforts, we created a well-labeled and sizable Shui manuscripts dataset, named Shuishu_T, which is the largest of its kind. Then, we applied target detection technology for Shui manuscript characters recognition. Specifically, we compared the advantages and disadvantages of Faster R-CNN, you only look once (YOLO), and single shot multibox detector (SSD), and subsequently chose Faster R-CNN to detect and recognize Shui manuscript characters. We trained and tested 111 classes of Shui manuscript characters with Faster R-CNN and achieved an average recognition rate of 87.8%. Finally, we designed a WeChat applet that can be used to quickly identify Shui manuscript characters in images obtained by scanning Shui manuscripts with a mobile phone. This work provides a basis for realizing the recognition of characters in Shui manuscripts on mobile terminals. Our research enables the intangible cultural heritage of the Shui people to be preserved, promoted, and shared, which is of great significance for the conservation and inheritance of Shui manuscripts.

**Keywords:** Shui manuscript; deep learning; target detection; dataset; character recognition





## 1. Introduction

The Shui nationality is one of 17 ethnic minorities in China with its own script. Shui manuscripts are ancient books bearing the ancient characters of the Shui people. The culture of the Shui manuscripts has been included in China's national intangible cultural heritage list since 2006. Shui manuscripts are rich in content, covering the knowledge of astronomy, geography, folklore, religion, and ethics of the Shui people. They are an important resources for understanding the unique culture of the Shui people and are of constructive significance for the study of history, anthropology, folklore, and even paleography. The Shui people refer to a person who is familiar with Shui manuscripts, known as Mr. Shuishu. Shui manuscripts are passed down from generation to generation through Mr. Shuishu, and the number of such persons is very small. This is because only the males of the tribe are eligible; they are not passed on to women or outsiders. At present, among the 410,000 Shui people, there are less than 200 Mr. Shuishus who can read Shui manuscripts, and the culture of Shui manuscripts is facing the threat of extinction.

The Chinese government completed the construction of the Shui manuscripts database on 19 July 2015, in which the pages of Shui manuscripts were scanned and the information saved in the form of images. Internationally, the British Museum in the UK has a project, called the "Endangered Archives Project," for which the aim is to create a multimedia and interdisciplinary database that will include digital images of Shui manuscripts, oral history

interview records, and audio and video recordings. At present, it contains 600 volumes of Shui manuscripts. However, these preservation methods for Shui manuscripts are very traditional and lack techniques for intelligent analysis of the data.

This study seeks to further strengthen the preservation of the intangible cultural heritage and explore new ways for the inheritance and development of the culture of Shui manuscripts. To this end, we applied artificial intelligence technology and used target detection methods to recognize Shui manuscript characters. Thus, this study makes the following contributions:

- We created a new image dataset of Shui manuscripts, named Shuishu_T. "Shuishu" is the English translation of the Chinese word "水书" and "T" is the first letter of the English word "Text," which indicates that the dataset is a text image dataset. This dataset is larger and contains more samples than any other known Shui manuscripts character dataset.
- We developed a target detection algorithm and applied it to the research on Shui manuscript characters recognition, and also trained a detection model for Shui manuscript characters.
- This research is conducive to the development of mobile applications, and we have realized it, allowing the intangible cultural heritage of Shui manuscripts to be inherited and promoted on mobile terminals.

## 2. Related Work

Optical character recognition technology is now quite mature and has been utilized for the detection and recognition of various mainstream languages in many fields, such as ticket recognition, intelligent medical procedures, and license plate recognition. However, character recognition of many endangered minority ancient manuscripts has not received sufficient attention. In recent years, various studies on text detection and recognition in some minority languages have been conducted, mostly using deep neural network-based detection methods. For example, Demilew and Sekeroglu [1] identified ancient Geez characters in Ethiopia and achieved 99.39% accuracy by using deep convolutional neural networks. Scius-Bertrand et al. [2] identified ancient Vietnamese manuscripts and proposed a learning-based alignment method combined with YOLOv5m for detection and recognition; they achieved an accuracy of 96.4% without manual annotation. Antony and Savitha [3] identified characters in images of Tulu palm leaf manuscripts and achieved a recognition rate of 79.92% using a deep convolutional neural network. Rahmati et al. [4] identified Persian characters and achieved an average accuracy of 99.69% on a dataset of five million Persian character images; they had the advantage of having access to an astonishing number of data samples. Vinotheni et al. [5] performed recognition of handwritten Tamil characters and proposed a modified convolutional neural network that recognized the characters with 97.07% accuracy on an experimental dataset.

Elkhayati and Elkettani [6] proposed a classification method combining computational geometry algorithms and convolutional neural networks for the recognition of handwritten Arabic characters. Han et al. [7] identified ancient Yi characters in China and proposed a detection method based on connected components and regression character segmentation, which solved the problem of character detection and segmentation of ancient_Yi characters. Liu et al. [8] identified oracle bones, the earliest text in China, and proposed using the target detection algorithm faster region-based convolutional neural network (Faster R-CNN) [9] combined with multi-scale fusion features to improve the text detection ability. They achieved an average accuracy rate of 88.2%. Gonçalves et al. [10] designed a Braille character detection system with an opportunistic deep neural network, which contributed to helping visually impaired people learn computer programming.

Only a few studies have been published on Shui manuscript characters recognition. Most studies are predominantly on Shui manuscript characters classification or characters segmentation. Yu and Xia [11] proposed a four-layer CNN that classified 50,000 Shui manuscript character images and achieved a classification accuracy of 93.3%. Xia [12] proposed an 11-layer CNN for Shui manuscript characters recognition and classified 60,000

Shui manuscript character images (including 80 categories). A classification accuracy of 98.3% was achieved. Zhao et al. [13] proposed a model that used the feedback of a CNN to determine the hyperparameters for clustering labels. Using 6,230 Shui manuscript character images in experiments, they demonstrated that the model which they proposed can be utilized for ancient text recognition. However, there were many errors in the text and automatic annotation. Ding [14] input a total of 8,500 Shui manuscript character samples from 17 categories into a CNN with only one layer and achieved a classification accuracy of 93.74%. Yang et al. [15] proposed a method based on adaptive image enhancement, region detection, and segmentation, and applied it to the segmentation of Shui manuscript character images. They achieved a good character segmentation effect.

## 3. Materials and Methods

### 3.1. Data Preparation

To create Shuishu_T, we visited libraries, museums, and areas where the Shui people live multiple times and also visited Shui manuscript experts. We collected a total of 1911 scanned images of Shui manuscripts from multiple volumes. The images of ancient Shui manuscripts are all stored in ".jpg" format with an average resolution of 1943 × 2924 pixels. The raw images of Shui manuscripts are not directly usable for experiments and require image correction, binarization, and noise reduction. In addition, the Shui manuscripts are very unusual and more difficult to read than other ethnic scripts, and only a very small number of Mr. Shuishu, who make up less than 5 in 10,000 people, can read them.

At present, there are more than 500 single characters in Shui manuscripts that can be read, and many characters exist in different volumes of Shui manuscripts, including more than 2000 characters in different forms. Various variants of the same type of character can also appear on the same page. As shown in Figure 1, this is a relatively common occurrence, where two variants of the character "jia" appear on the same page of a Shui manuscript. Consequently, it is necessary to communicate with experts for confirmation when labeling, and it is not possible to hand over to a labeling company for bulk labeling. This makes it extremely difficult to create a dataset.

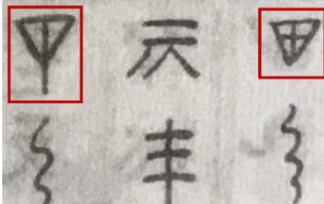

**Figure 1.** Example of Shui manuscript characters.

After manual labeling, 1734 images with labels were obtained, comprising 111 categories and a total of 91,127 characters. Statistics on the labeled Shui manuscript images show that the number of samples in each category is extremely unbalanced. Among them, the category with the smallest number of samples is "ĬĬ", with only one sample; the category with the largest number of samples is "Ŧ", with 7524 samples. Some categories had very small sample sizes that could not support the training of the model and required data augmentation. For classes with small sample sizes, we used several data augmentation methods, such as manual handwriting, image synthesis, and image cropping.

First, manual handwriting data amplification was performed for the category of Shui manuscripts with a sample size of less than 300, and 51 handwritten Shui manuscript images were obtained with a total of 12,651 character samples after 20 handwriting amplifications by different volunteers. Although the manual handwriting amplification method is simple, it has the disadvantages of low efficiency and high cost. Figure 2 shows the image obtained via the manual handwriting method, and Table 1 compares the sample size before and after handwriting amplification in some categories.

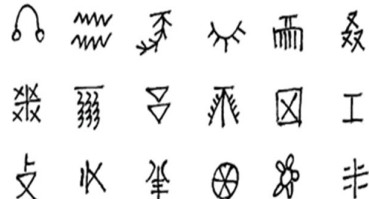

**Figure 2.** Image obtained via the manual handwriting method.

**Table 1.** Comparison of sample size before and after handwriting amplification in some categories.

| Category | Glyph | Meaning | Original Quantity | Number of Handwriting |
|----------|-------|---------|-------------------|------------------------|
| yint | | cloudy | 2 | 446 |
| jiuh | | nine fires | 11 | 539 |
| xiang | | incense | 1 | 522 |
| bo | | monogram | 28 | 520 |
| chunt | | spring | 36 | 475 |
| shuz | | comb | 6 | 471 |
| spo | | slope | 18 | 453 |
| ceng | | layer | 34 | 452 |
| jih1 | | taboo | 94 | 428 |

Second, for target samples that would not be ambiguous after orientation transformation, we used image synthesis to augment the data. A single character slicing operation was performed on the 60 classes with a sample size of less than 500 in Shui manuscript images. Subsequently, binarization, rotation, flipping, and noise addition were performed on the sliced characters. Finally, the sliced characters were randomly combined. Through the image synthesis operation, we obtained 1600 images with an average size of 1200 × 1800 pixels, containing 38,383 character samples. These character samples contained various scenes, such as different text orientations or various noises. The data are thus diverse, which can enhance the generalization ability of the model and prevent overfitting. Figure 3 shows the image obtained via the image synthesis method.

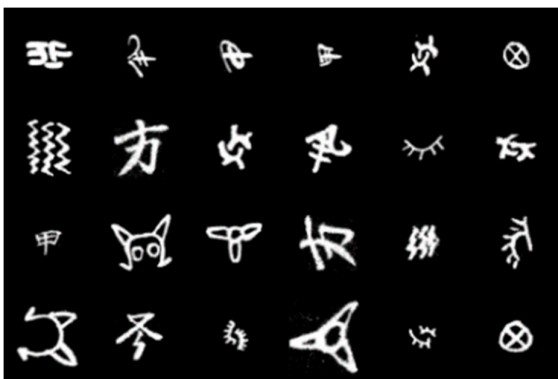

**Figure 3.** Image obtained via the image synthesis method.

Third, we cropped 1734 original images and 51 images added by manual handwriting, using the top, bottom, left, right, and center positions as loci of action. With the cropping method, we obtained 10,583 Shui manuscript images with an average size of 1002 × 1509 pixels, containing 126,406 character samples. Figure 4 shows the image obtained via the image cropping method.

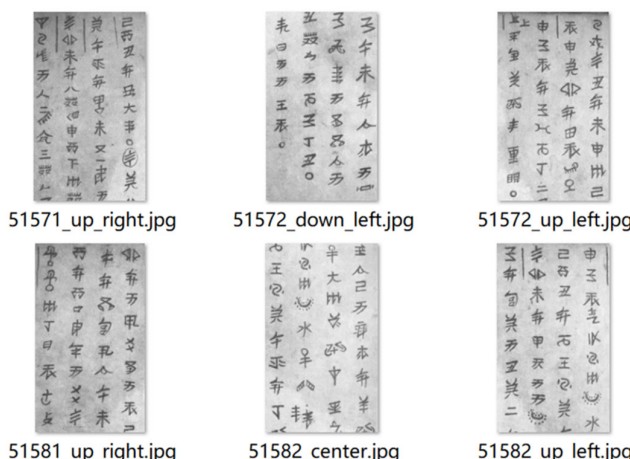

**Figure 4.** Image obtained via the image cropping method.

Finally, the well-labeled and sizable Shuishu_T contained 10,583 text images with labels, with the average size of the images being 1002 × 1509 pixels, for a total of 164,789 samples of Shui manuscript characters in 111 categories. Among them, there are seven categories with fewer than 500 samples, accounting for 6%; a further 37 categories with 500 to 1000 samples, accounting for 33%; and 67 categories with more than 1000 samples, accounting for 61%. As shown in Table 2, Shuishu_T is larger and contains more samples than the known Shui manuscripts character datasets. We discuss the effectiveness of our proposed data enhancement method through ablation experiments in Section 4.3.

**Table 2.** Comparison of other datasets of same class.

| Dataset | Number of Categories | Number of Character Samples |
|---|---|---|
| Yu and Xia, 2019 | Unknown | 60,000 |
| Zhao et al., 2020 | Unknown | 6230 |
| Ding, 2020 | 17 | 8500 |
| Shuishu_T (our) | 111 | 164,789 |

*3.2. Target Detection Models*

The aim of target detection is to detect whether an image contains certain targets and to identify and locate those targets accurately; it is an application direction in the field of computer vision. The most popular target detection algorithms can be divided into two classes. The first class of algorithms is called "two-step" detection algorithms, in which a dense set of candidate regions is first sampled on the feature map, and then the candidate regions are classified and regressed with high detection accuracy, typically by region-based CNN (R-CNN) [16], Fast R-CNN [17], or Faster R-CNN. The other class of algorithms is called "one-step" detection; i.e., direct sampling and regression on a multi-layer feature map, generating the class probability and location coordinate values of the object and outputting them in a single step, which is fast. Typical algorithms include you only look once (YOLO) [18] and single shot multibox detector (SSD) [19].

R-CNN, proposed by Girshick et al. in 2014, is one of the first algorithms to apply deep learning techniques to target detection. The data features of the candidate regions are synthesized and classified using a support vector machine. Compared with some traditional algorithms for target detection, R-CNN replaces the feature extraction part with a deep neural network, which is a breakthrough in terms of accuracy and speed. However, the method is computationally overloaded and trades off a large amount of resources and runtime for improved accuracy.

Fast R-CNN is an upgraded version of R-CNN that adds a region of interest (ROI) pooling layer after the last convolutional layer of the CNN. This allows each input image of the network to be of arbitrary size and only one feature extraction for each image, greatly

improving efficiency. In addition, Fast R-CNN uses Softmax instead of support vector machines for multi-task classification, which significantly improves the target detection efficiency again. However, it has the problem of time-consuming selection of candidate regions.

The main feature of Faster R-CNN, published in 2017, is the change of the candidate region extraction method, i.e., the use of the region proposal network (RPN), which allows CNNs to generate candidate regions directly, instead of the selective search method used previously. The RPN can automatically extract candidate regions quickly and efficiently, and the alternating training method of RPN and Fast R-CNN can improve the accuracy of target detection while greatly reducing the time spent on target detection. The overall process can be categorized into four steps: candidate region generation, feature extraction, classification, and position refinement. The overall structure of the Faster R-CNN is shown in Figure 5.

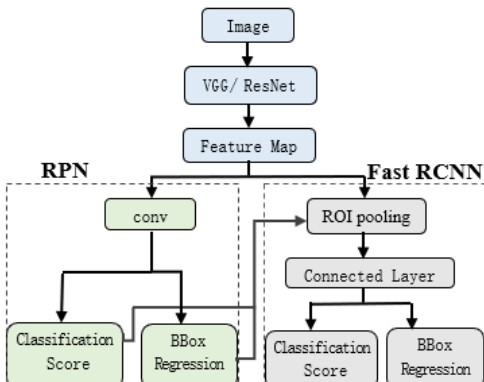

**Figure 5.** The overall structure of the Faster R-CNN.

The core idea of YOLO is to solve the object detection problem as a regression problem, using the whole image as the input to the network, and training and detecting in one network throughout. Published in 2016, YOLO has undergone multiple generations: YOLOv1, YOLOv2 [20], YOLOv3 [21], YOLOv4 [22], and YOLOv5. YOLOv1 directly predicts candidate frame locations, creatively combining recognition and localization into one. YOLOv2 uses offset prediction to speed up the convergence of the network. YOLOv3 provides two major improvements, namely, the use of residual models to deepen the backbone network and the use of the feature pyramid network (FPN) architecture for multi-scale detection. YOLOv4 uses cross stage partial darknet (CSP-Darknet) as the backbone network, which reduces model parameters, enhances rich feature information with Mosaic data, increases the number of positive samples with neighborhood positive sample candidate frame matching calculation, and converges the network more. YOLOv5 enhances the feature fusion capability based on YOLOv4 with the HardWish activation function and provides more flexible parameter configuration to improve the small object detection performance.

SSD, published in 2016, combines the candidate frame mechanism and regression ideas of Faster R-CNN and YOLO to regress different features at multiple scales of regions at different locations in images, extract and analyze these features in layers, and perform computational operations such as scale regression and feature classification of borders in turn. Ultimately, it completes the training and detection tasks for targets in multiple regions of different scales. The SSD algorithm improves the target detection accuracy without affecting the speed. However, because the default box shape and grid size of this algorithm are predetermined, the detection effect for small targets is still unsatisfactory.

## 4. Experimental Evaluation

### 4.1. Experimental Environment

Experiments and evaluations were performed in a deep learning environment with a TensorFlow backend. The hardware platform comprised a NVIDIA GeForce GTX 2080 Ti graphics card. The experimental codes were all written in Python. The environment was the same for all of the experiments.

### 4.2. Performance Evaluation

In the experiment of object detection, in terms of recognition accuracy, the mean average precision (mAP) was used as the evaluation metric and mAP is used to evaluate the quality of the model, an important assessment metric in target detection, which is defined as follow:

$$P_{mAP} = \frac{\sum_{i=1}^{N} P_i^A}{N} \times 100\% \tag{1}$$

where $N$ is the number of categories and $P_i^A$ is the accuracy of category $i$, which is defined as follows:

$$P_i^A = \sum_{i=1}^{N} P(i)(r_i - r_{i-1}) \tag{2}$$

where $P(i)$ is the precision for the category $i$ and $(r_i - r_{i-1})$ is the change in recall for category $i$.

### 4.3. Model Comparison Experiments

#### 4.3.1. Experimental Setup

To select a suitable model for Shui manuscript character detection, we conducted two sets of experiments using different combinations of Shui manuscript image datasets as input. Because the image size required by the SSD is 300 × 300 pixels, we need to crop the large image to this size. Therefore, Experiment 1 and Experiment 2 use different datasets. Table 3 shows the dataset information for the two sets of experiments, and each dataset has a large number of text images, and each image contains many samples of Shui characters. The dataset used in Experiment 1 contains 60,124 Shui manuscript images, all without data enhancement. The images had 81,371 character samples in 114 categories and were all in ".jpg" format with a size of 300 × 300 pixels. The dataset used in Experiment 2 was Shuishu_T introduced in Section 3.1, which contained 10,853 Shui manuscript images. The images had 164,789 character samples in 111 categories and were all in ".jpg" format with a size of 1002 × 1509 pixels. To ensure a consistent data distribution, we used the hold-out method to divide the dataset. We followed the usual practice of dividing the training and test sets in a 7:3 ratio and selecting a small number of samples from the training set to form the validation set.

**Table 3.** Datasets information for the two sets of experiments.

| Experimental Group | Dataset Size | | |
| --- | --- | --- | --- |
| | Number of Categories | Number of Images | Number of Character Samples |
| Experiment 1 | 114 | 60,124 | 81,371 |
| Experiment 2 | 111 | 10,583 | 164,789 |

#### 4.3.2. Results and Analysis

In the first set of experiments, we fed the same dataset into three models—Faster R-CNN, YOLO, and SSD—for training and testing. The experimental results are listed in Table 4.

**Table 4.** Results of the first set of experiments.

| Model | mAP (%) |
|---|---|
| Faster R-CNN | 76.7 |
| YOLOv5 | 79.2 |
| SSD | 25.4 |

YOLOv5 achieved the highest mAP of 79.25%. The mAP of Faster R-CNN was slightly lower than that of YOLOv5, and that of SSD was much lower than that of the first two models. In the image testing part, it was found that Faster R-CNN could recognize the characters in the images better, whereas YOLOv5 could not fully recognize the Shui manuscript characters of the small targets and its misrecognition was more serious. SSD could not recognize the small target characters in most of the test images owing to its low accuracy. As shown in Figure 6, Faster R-CNN was able to accurately detect the complete characters in the same image and recognize them correctly. Although YOLOv5 also detected three characters, it only correctly detected two of the complete characters, missed one complete character, and incorrectly detected one incomplete character. SSD only detected one character in the image, confirming that SSD is less effective for detecting small and multi-target Shui manuscript text images.

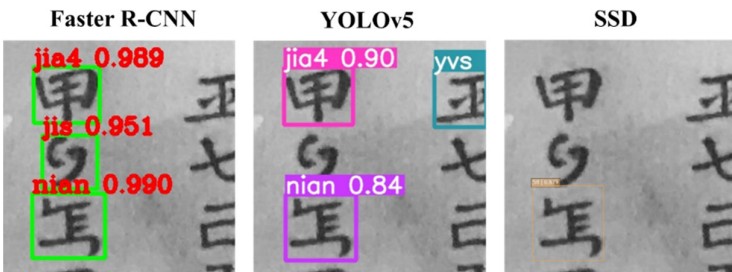

**Figure 6.** Image test results of Faster R-CNN, YOLOv5, and SSD.

Because of the poor detection performance of small and multitarget images by SSD, we only compared YOLO and Faster R-CNN in the second set of experiments. We utilized the Shuishu_T dataset with target detection models: YOLOv5s, YOLOv5x, and Faster R-CNN. YOLOv5s and YOLOv5x are the latest YOLO series models, with the model size of YOLOv5s being much smaller than that of YOLOv5x. It is well known that the role of the backbone network is to extract the overall features of the input image, which is the key to the target detection method. We used two types of CNNs, VGG-16 [23] and ResNet-101 [24] as the backbone of the Faster R-CNN because they obtained satisfactory accuracies, both above 99%, in the classification experiments of Shui characters. This allowed us to compare the performance of different backbone networks in the "two-step" detection algorithm. VGG-16 is an efficient and straightforward deep CNN focused on building convolutional layers. It contains 16 convolutional layers and fully connected layers, which is relatively regular. Usually, each set of convolutional layers is followed by a pooling layer that can compress the image size. ResNet-101 is a residual network with 101 layers and a residual block. The residual block is often called a jump layer connection—that is, the output of some layers skips one or more layers directly so that the information is passed to the deeper layer of the neural network and solves the problem of network degradation caused by network deepening. Intersection over union (IOU) is the ratio of the intersection and union between the "predicted area" and the "real area," and it is also a measure of object detection accuracy. For the tests, we calculated and averaged mAP values with IOU values equal to 0.5, 0.6, 0.7, 0.8, and 0.9. The results are shown in Table 5.

**Table 5.** Results of the second set of experiments.

| Model | Backbone Network | mAP (%) |
|---|---|---|
| YOLOv5s | CSP-Darknet | 80.3 |
| YOLOv5x | CSP-Darknet | 84.1 |
| Faster R-CNN | VGG-16 | 85.8 |
| Faster R-CNN | ResNet-101 | 87.8 |

The results show that Faster R-CNN achieved a higher mAP than that of YOLOv5 on the test set. The Faster R-CNN model with ResNet101 as the backbone network always achieved a better mAP than the model with VGG16 as the backbone network, which achieved a maximum mAP of 87.8%. Based on the experience of the first set of experiments, the recognition rate of YOLOv5 was high, but the test results of images were not superb, hence, we continued with the image test. Figure 7 shows an example of the test images.

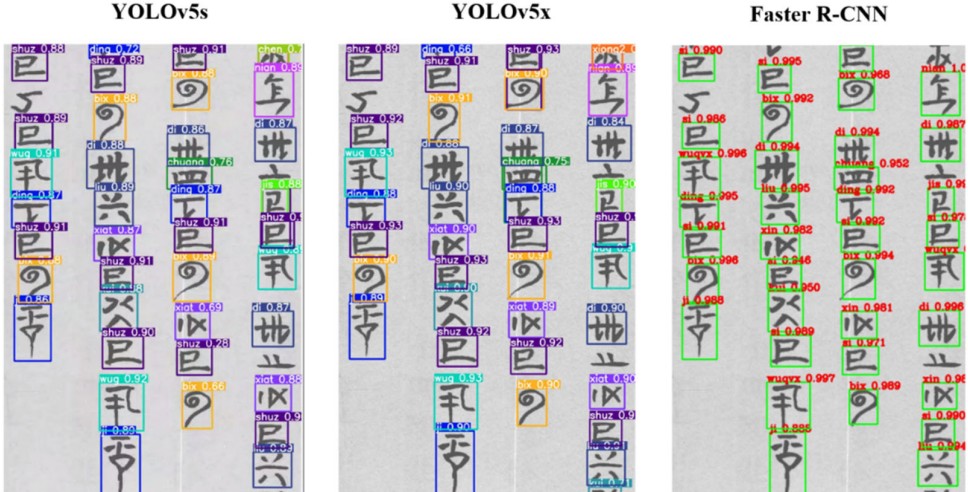

**Figure 7.** Image test results of YOLOv5s, YOLOv5x, and Faster R-CNN.

It was found that all three methods could detect the text in the ancient images of the water book well, but the YOLO model still had more misrecognition problems. For example, the test images of YOLOv5s and YOLOv5x both misidentified two classes of characters, misidentifying all the "si" as "shuz" and all the "wuqvx" were incorrectly identified as "wug". In addition, YOLOv5x also misidentified "chen" as "xiong2", whereas Faster R-CNN recognized all the words in the test images correctly. According to our analysis, the error rate of YOLOv5 is due to the use of two data augmentation techniques in the YOLOv5 model—one is Mosaic and the other is random erasure. These two methods contribute greatly to the detection of person, animals, or other objects, but are not applicable to Shui characters. Because human characters have their own specificity, when some features of characters are erased, they may lose their original meanings and become new characters, so there is a high possibility of misclassification.

Shui manuscript characters recognition is a small-target and multi-target detection problem, with the focus being on whether the characters in Shui manuscripts can be accurately detected and recognized. The above experimental results show that YOLO and SSD have many problems detecting small and multiple targets. Thus, Faster R-CNN is the most suitable for the detection and recognition of Shui manuscript characters. Furthermore, in 2018, Julca-Aguilar and Hirata [25] proposed the use of Faster R-CNN as a method for detecting symbols in handwritten graphics, and experimentally evaluated it on graphical and mathematical expression datasets. Their results showed that Faster R-CNN can be effectively used for handwritten symbol recognition. In 2019, Yang et al. [26] proposed an improved Faster R-CNN algorithm for natural scene text detection. They reported

experimental results demonstrating that it improved the detection of small target characters. Considering the successful application of Faster R-CNN in text detection reported by these two studies and the results of our evaluation experiments, we chose Faster R-CNN to train the Shui manuscript characters recognition model and applied it to the development of the mobile applet.

### 4.4. Ablation Experiments

#### 4.4.1. Experimental Setup

To verify the effectiveness of the data enhancement method proposed in this study, we carried out ablation experiments by inputting different combinations of Shui manuscript datasets into Faster R-CNN. All the datasets were divided into training set and test set in a ratio of 7:3, and a small number of samples in the training set were selected to form the validation set. Table 6 shows the dataset information for each experiment.

**Table 6.** Dataset information of each experiment.

| Experimental Group | Dataset Combination | Number of Images | Number of Character Samples |
|---|---|---|---|
| Experiment 1 | raw data | 1734 | 91,127 |
| Experiment 2 | raw data + handwritten data | 1773 | 103,778 |
| Experiment 3 | raw data + handwritten data + image composite data | 83373 | 142,161 |
| Experiment 4 | Shuishu_T (cropped "raw + handwritten" data + image composite data) | 10583 | 164,789 |

The initial learning rate was set to 0.001, the training iterations were 60,000, and the learning rate decayed to 0.0001 after 10,000 iterations. The momentum was set to 0.9. The batch size of RPN was set to 512.

#### 4.4.2. Results and Analysis

The training loss of each experiment is shown in Figure 8. It can be seen that each experiment converged faster, and after training to 30,000 rounds, the change in the loss decreases and the curve tends to be smooth, with the least oscillation occurring in Experiment 4, i.e., the experiment with the input being the Shuishu_T dataset.

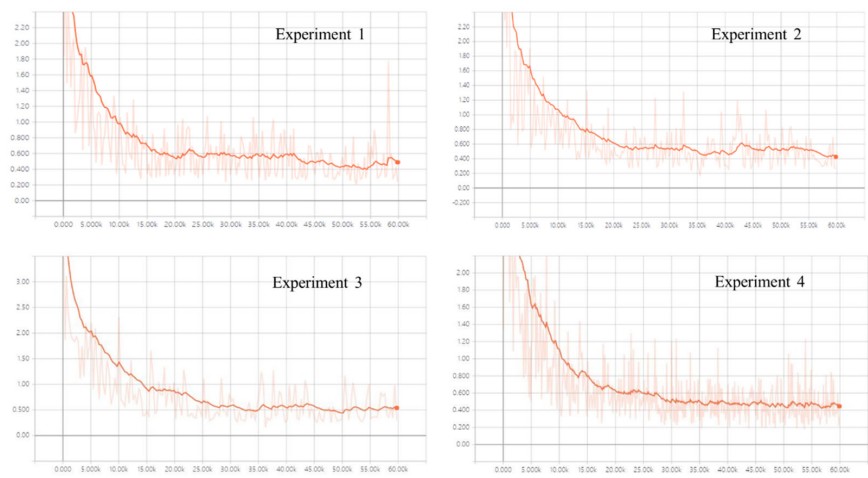

**Figure 8.** Training loss of each experiment.

The test results of each experiment on the test set are shown in Table 7. The results show that the mAP of Experiment 1 was not high, only 67.4%. The highest mAP was 99.4% and the lowest was zero. There were 13 categories that were not recognized, thus lowering the average recognition accuracy—indicating that the crux of the low mAP lies in the dataset rather than the model.

**Table 7.** Results of each experiment.

| Experimental Group | Sample Size | mAP (%) |
|---|---|---|
| Experiment 1 | 91,127 | 67.4 |
| Experiment 2 | 103,778 | 68.8 |
| Experiment 3 | 142,161 | 78.9 |
| Experiment 4 | 164,789 | 85.8 |

In Experiment 2, manual handwriting data were added and a mAP of 68.8% was obtained, an improvement of 1.4 percentage points over Experiment 1. The number of categories in Shui manuscripts that could not be recognized was reduced to 11. The reason there were still target samples that could not be recognized is that the amount of manually handwritten data was still small. Although the improvement in mAP for Experiment 2 was small, it proved that the manual handwriting approach to data augmentation can effectively improve recognition accuracy.

In Experiment 3, data from image synthesis were added, resulting in a further increase in the amount of data for the Shui character samples and a gradual reduction in the gap between the categories of rare characters and commonly used characters. Thus, in Experiment 3, a mAP of 78.9% was obtained, an increase of 10.1 percentage points over Experiment 2. In Experiment 3, all 111 categories of Shui characters were recognized. The highest mAP obtained for each category in Experiment 3 was 99.8%, and the lowest mAP was 18.2%. The significant increase in mAP in Experiment 3 is due to the addition of 38,383 samples of rare characters by image synthesis. In addition, the diversity of character samples added by image synthesis enhanced the generalization ability of the model, which is also one of the reasons for the improved recognition accuracy.

In Experiment 4, the Shuishu_T dataset with cropped images was used, and it had 22,628 more character samples than in experiment 3. In the end, in Experiment 4, a mAP of 85.8% was obtained, which is 6.9 percentage points higher than that in Experiment 3. Experiment 4 demonstrates that image cropping is also effective in improving the recognition accuracy of characters in Shui manuscripts.

## 5. Application of the Shui Manuscript Characters Recognition Model

### 5.1. Project Introduction

To explore new ways of preserving and developing the intangible cultural heritage of Shui manuscripts and to enable more people to easily and quickly access, use, and spread the culture of Shui manuscripts, we applied the Shui manuscript characters recognition model trained by Faster R-CNN to the development of a mobile WeChat applet. We designed and implemented the Shui characters recognition application applet. Figure 9 shows the development process of the Shui characters recognition applet.

The Shui characters recognition applet is a text-recognition application for cultural promotion. It contains a Shui Characters Recognition Module, Cultural Encyclopedia Module, and Developer Introduction Module. The Shui Characters Recognition Module comprises two parts: Shui manuscripts dataset and handwritten Shui characters recognition. The handwritten Shui characters recognition is the core function. It allows the user to take a picture or select an image from a photo album and submit it by clicking the "submit recognition" button. When this is done, the system uploads the image to the server for recognition and returns the result to the user, who can then choose to save the result image locally, re-identify it, or return it. The Cultural Encyclopedia Module provides users with an understanding of the Shui people and the culture of Shui manuscripts. The Developer Introduction Module introduces the research content, objectives, and team members.

The WeChat applet is a front-end program that primarily employs JSBridge to allow the information between the upper layer development and the system layer to be passed through the API interface to achieve the call to the lower layer. To get the project online and operational, we rented a 4-core, 8 GB lightweight application server provided by Tencent Cloud. The program ran on a server with an Intel(R)® Xeon®(R) Platinum 8255C, 2.50

GHz CPU. The average inference time for a user to submit a Shui manuscript characters detection task was 6.59 s.

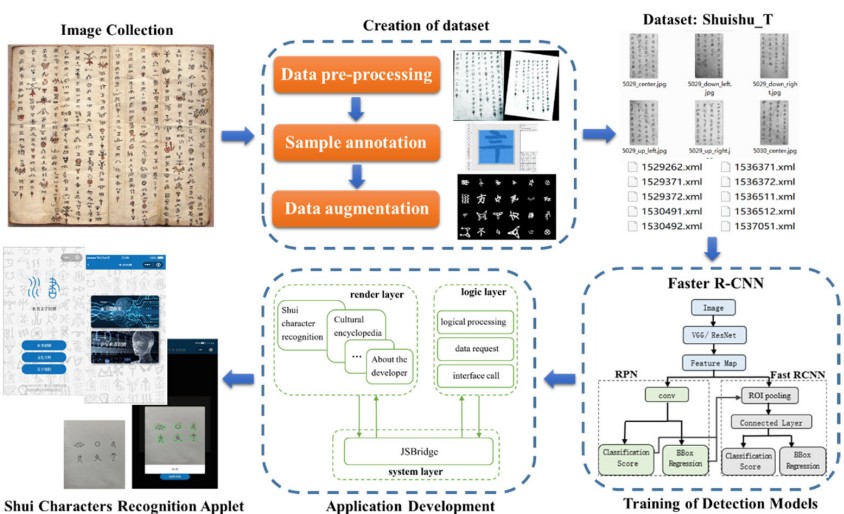

**Figure 9.** Development process of the Shui characters recognition applet.

### 5.2. Real Device Test

The Shui characters recognition applet was developed for online operation. To test the effectiveness of its core functionality, we invited eight volunteers to use the applet. Five of the volunteers had mobile phones running Android, whereas the other three had mobile phones running iOS. After the test, all the volunteers said that they could successfully use the applet to detect and recognize Shui characters. A volunteer wrote some Shui characters on a piece of paper, then took a photo and uploaded it for recognition. As shown in Figure 10, the left side is the test image written on the paper before recognition, and the right side is the submitted recognition result. We found that all six Shui characters were detected and recognized correctly, which is consistent with our development goals. We will continue to maintain and improve this application in the future, and thereby achieve the preservation, inheritance, and promotion of the culture of Shui manuscripts.

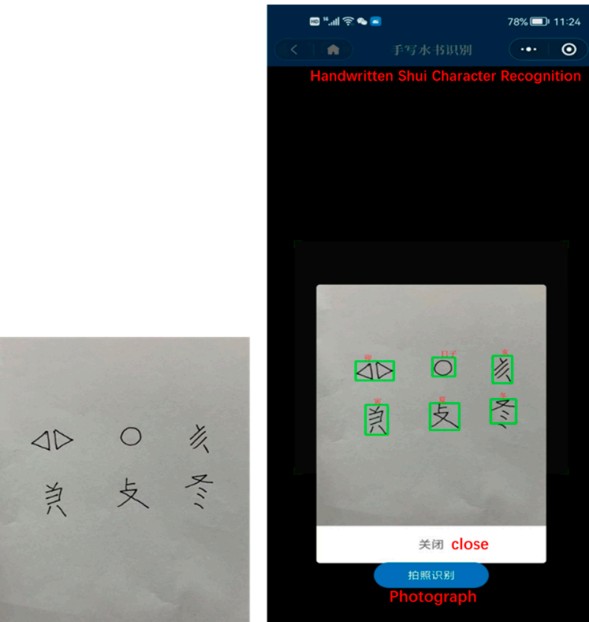

**Figure 10.** Application of Shui characters recognition module.

## 6. Conclusions

Shui manuscripts provide a historical testimony of the national identity and spirit of the Shui people. In this study, in response to the lack of a high-quality Shui manuscript datasets, we collected Shui manuscript images in the Shui area and used various methods to enhance them. Through our efforts, we created a well-labeled and sizable Shui manuscripts dataset, named Shuishu_T, which is the largest such dataset. Then, we applied target detection technology for Shui manuscript characters recognition. Following comparison of the advantages and disadvantages of Faster R-CNN, YOLO, and SSD, we chose Faster R-CNN to train the Shui manuscript characters recognition model. We trained and tested 111 classes of Shui manuscript characters with Faster R-CNN and achieved an average recognition rate of 87.8%. The model can accurately locate and recognize Shui manuscript characters in test images. This work provides a basis for realizing the recognition of characters in Shui manuscripts on mobile terminals. Finally, we designed a WeChat applet that users can quickly utilize to identify Shui manuscript characters in images obtained by scanning Shui manuscript images with their mobile phones. This enables the intangible cultural heritage of the Shui people to be promoted in the mobile arena. In the future, for the Shui manuscripts, our research will focus on machine translation, speech recognition, and character recognition of complex typeset text. Our research is of great significance for the protection, inheritance, and promotion of the cultural heritage of Shui manuscripts in the new era. Furthermore, this study provides ideas for the inheritance and development of other minority languages and cultures.

**Author Contributions:** Conceptualization, M.T. and X.L.; methodology, M.T. and X.L.; software, S.X., M.H. and M.T.; validation, S.X., M.H. and M.T.; formal analysis, M.T. and X.L.; investigation, M.T. and X.L.; resources, M.T. and X.L.; data curation, S.X., M.H. and M.T.; writing—original draft preparation, M.T. and S.X.; writing—review and editing, M.T. and X.L.; visualization, S.X., M.H. and M.T.; supervision, M.T. and X.L.; project administration, M.T. and X.L. All authors have read and agreed to the published version of the manuscript.

**Funding:** This research received no external funding.

**Institutional Review Board Statement:** Not applicable.

**Informed Consent Statement:** Informed consent was obtained from all subjects involved in the study.

**Data Availability Statement:** Please scan the QR code below to enter the WeChat applet of the Shui characters recognition application.

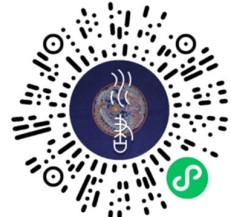

**Conflicts of Interest:** The authors declare no conflict of interest.

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
