# Peer review of "Character Recognition in Endangered Archives: Shui Manuscripts Dataset, Detection and Application Realization"

_applsci, doi:10.3390/app12115361_

Round 1

Reviewer 1 Report

1) The study is judged to have a higher industrial meaning than an academic meaning

2) Data collection and database creation are well worthwhile, but more detailed description of the database is required

2-1) Table descriptions such as the number and amount of databases and the type of labels should be added

2-2) It appears that the shui character has been used in the study in the related study, then it is necessary to explain what is better than the existing database 

3) The target detection model is already a well-known model and needs to be written more briefly

4) Using deep learning for shui character detection has no academic significance.  I think it is necessary to change the deep learning model that is more specialized in shui characters

5) Write in detail about the experimental method

5-1) The experiment was divided into three parts, but I don't know why. It is necessary to write the criteria for dividing the datasets of the experiment

5-2) Are the experimental environments all the same? (Deep learning framework, computer specifications, etc.)

6) This paper feels more like a business report than a paper. The final result, the character recognition module, is also insufficient in academic value and has a strong industrial character.

7) Recommendations on developing deep learning models that are more specialized in shui characters

Author Response

Dear reviewer,thank you very much for your suggestions.Please see the attachment.

Reviewer 2 Report

The dataset is well described and its new dataset for understanding Chinese culture.

The author must explain the similarities with other dataset of same class

Formatting and grammatical issues must be resolved. Many spellings mistakes were noticed it is recommended to get spell check.

No novel approach is presented. 

Author must highlight its contribution in more crisp manner.

Classwise accuracies must be explained and also few misclassification must also be discussed.

  • Comparative analysis of proposed dataset with currently available dataset.
  • Highlight the image enhancement approaches used and show the respective results.
  • The authors have used YOLO,SSD and R-CNN for target detection. it is required to mention why authors opted only for these approaches.
  • The above-mentioned approaches must be applied to other similar datasets(available in the literature) to highlight the importance of these approaches.
  • The main contribution needs improvement. you can break it into more steps to exactly highlight the main contribution.
  • You have achieved multiple milestones and you should write all in separate bullets and each contribution should be seen in the results section.
  • The author must add a few citations for previously available datasets and why there is a need for new one.
  • Figure 2 needs to be more visible.
  • The experimentation is very nicely presented. Add class labels details and also how authors have used YOLO, SSD and R-CNNs.
  • Add AuC/ROC curve to enhance your results section.
  • Figure and caption must be on the same page.

Author Response

Dear reviever, thank you very much for your suggestions. Please see the attachment.

Reviewer 3 Report

This research develops a dataset to detect Shui characters using object-detection techniques. Specifically, the authors carry out experiments using YOLOv5 (its variants such as YOLOv5s, YOLOv5x, etc.), Faster R-CNN and SSD.

Faster R-CNN is finally chosen to be the most appropriate network for the recognition and classification of Shui characters.

The authors create a WeChat app, which can be used to detect and classify Shui characters from the image.

+ The manuscript is pretty easy to understand.

+ A large dataset is developed for the recognition of Shui dataset.

+ This paper demonstrates a good application of object detection algorithm.

- The authors say that the image is uploaded to the server for the recognition of Shui characters, however they do not provide any details of the server. Does it use a GPU, or the inference is carried out on a CPU? How much time does it take to carry out the inference? Provide some benchmark and details.

- Authors use an old backbone ResNet-101, which is pretty outdated now. The common trend in 2020s is either to use transformer-based network (Swin-T) or the recent CNN ConvNext. If feasible, it would be better to include another backbone e.g., ConvNext for evaluation.

Minor comments:

Multiple space between “which” and “is” on line 24;

Case is not consistent in “Keywords”

Multiple spaces on Line 250

“a” should be capital in “although” on line 258

Line 422, fix space between “.” and “Our”

Also, check others by yourself.

Round 2

Reviewer 1 Report

1. I checked all your major revision. I think most of the comments have been revised.

2. What is the difference between Experiment 1 and Experiment 2 in Table 3? What is the difference between 114 and 111?

3. It is difficult to understand the difference between the number of images and the number of samples. Please clearly indicate the difference between the two. 

4. I think it's worth researching in terms of application, application development, and database accumulation. Do you plan to release the database for future research?

Reviewer 2 Report

The comments are incorporated in the revised version

Author Response

Dear reviewer, thanks for your hard work. Please see the attachment.

Reviewer 3 Report

The authors have done a good job in revising the manuscript and also providing valid responses to me as well as to other reviewers. I do not see any need for introducing a new network for Shui character recognition as this Journal is about application-based research and provided that the existing accuracy obtained by the researchers is pretty good. Improving overall detection accuracy in future work is recommended.
Thank you!
